# Can Media Attention Promote Green Innovation of Chinese Enterprises? Regulatory Effect of Environmental Regulation and Green Finance

**Fengyan Wang** [1,2,3] , **Ziyuan Sun** [1,3,*] **and Hua Feng** [2]

1   School of Public Policy & Management, China University of Mining and Technology, Xuzhou 221116, China
2   School of Accounting, Shandong Women's University, Jinan 250300, China
3   School of Economic and Management, China University of Mining and Technology, Xuzhou 221116, China
*   Correspondence: 4236@cumt.edu.cn

**Abstract:** Under the green sustainable development strategy, media attention has played a vital role in promoting green innovation of enterprises. Nevertheless, existing research mainly analyzes how media attention affects enterprise innovation behavior, while ignoring the role of environmental regulation and green financial policy. The main purpose of this study is to explore whether the media's attention to environmental issues can promote the enterprise's green innovation, whether the constraint policies of environmental regulatory and incentive policy of green financial can play regulatory roles, and whether these effects are heterogeneous among different types of enterprises. Based on the data of Chinese A-shared listed enterprises from 2010 to 2019, this paper draws the following conclusions by constructing the fixed effect models: First, media attention promotes the substantive and strategic green innovation of enterprises. Second, environmental regulation and green finance have positive regulatory effects on media attention and enterprise green innovation, and these regulation effects have a more significant impact on promoting substantive green innovation. Third, media attention is more sensitive in heavy pollution and state-owned enterprises green innovation. The results of the robustness test still support the conclusion, indicating that the conclusions are reliable. The research of this paper helps to clarify the role of environmental regulation and green finance in media attention and green innovation on a deeper level and puts forward targeted suggestions on how to stimulate enterprise green innovation from the perspective of media, government, and enterprises.

**Keywords:** media attention; environmental regulation; green finance; green innovation

## 1. Introduction

With the advent of the era of big data, the speed of information transmission has been greatly accelerated, and the new media that mainly disseminate information are exhibiting a development trend of diversification, intelligence, and integration [1,2]. As a form of informal supervision, media attention acts as a social supervisor, becoming an effective supplement to the government supervisory system [3], and plays an essential role in the era of information technology and media integration [1,4]. Media attention has gradually become a key factor influencing the reputation and profitability of enterprises in the new media era [5].

The Chinese government proposed to establish a green and low-carbon economic system for circular development since 2021, ensure the realization of the goals of carbon peak and carbon neutrality, and promote green development to a new level. The China's Government Work Report for 2022 proposed that the ecological environment must be continuously improved to elevate green and low-carbon development level. This shows that green sustainable development has risen from the enterprise level to a national strategy [6]. Under the strategy of low-carbon development, how to coordinate the balance between

economic growth and environmental protection has also attracted extensive attention of the international community [7,8]. Green innovation is a crucial driving force to realize the green development of enterprises and the sustainable development of the regional economy [9,10]. It is also an indispensable measure to motivate economic growth and high-quality development while realizing environmental protection [11]. Accordingly, the environmental behavior and green innovation activities of enterprises have also attracted more attention from the media [12]. Nevertheless, media attention may not only have a positive role in green governance from the perspective of effect [13,14], but also may lead to negative "greenwashing" behavior [15,16]. Many studies show that media reports can bring out the public's attention to the issue of environmental responsibility of enterprises [17,18]. Media attention may have a crucial impact on corporate reputation [19]. Correspondingly, the government will take measures to restrict the environmental behavior of enterprises and encourage them to ameliorate the reputation through green innovation [4,14]. Media reports on environmental issues will cause stakeholders to worry about the sustainable development of enterprises [20], form the legitimate pressure of stakeholders [21,22], and enable enterprises to raise environmental performance and change their image in public through green innovation. Nevertheless, due to the shortcomings of high-risk and long-term benefits of green innovation activities, it is also possible that enterprises will implement "greenwashing" measures [15], such as emissions fraud and false publicity, which are not conducive to substantial green innovation [16].

Existing studies have explored how media attention affects enterprise green innovation from positive [13,14] and negative perspectives [15,16]. However, these conclusions have not reached an agreement, which provides a starting point for this study. In addition, there are still some problems that have not been discussed in the current research. For example, whether there are some differences in the impact of media attention on substantive green innovation and strategic green innovation of enterprises, and whether the constraint policy of environmental supervision and the incentive policy of green finance have heterogeneous effects on the micro behavior of enterprises under the pressure of media attention. Furthermore, whether there are some differences in the impact of media attention on green innovation of different types of enterprises. Consequently, different from the current research, this paper will further explore whether media attention will enhance the green innovation of Chinese enterprises, reveal whether environmental regulation and green finance play a regulatory role between media attention and green innovation, and verify whether these effects are different among different types of enterprises. The answer to the above questions will help us accurately understand the role of media attention on the green innovation behavior of enterprises, and then help government departments better formulate targeted policies to encourage enterprises to accelerate green innovation.

In order to deeply explore the relationship between media attention and enterprise green innovation, as well as the effects of environmental regulation and green finance, this paper uses the data of Chinese A-shared listed enterprises from 2010 to 2019 to empirically test the impact of media attention on enterprise green innovation by establishing the time and province dual fixed effect model. The empirical test results found that the media attention improves the green innovation of Chinese enterprises significantly, and both substantive and strategic green innovation have incentive impact. From the perspective of regulatory effect, environmental regulation and green finance play the positive regulatory role between media attention and enterprise green innovation, while regulatory effect is more sensitive in promoting substantive green innovation. In addition, media attention has a more significant impact on heavy pollution and green innovation of state-owned enterprises. The robustness test by changing variables and the endogenous test still supports these conclusions. From the practice, improving the green innovation vitality and innovation quality of enterprises is a pivotal measure to improve high-quality economic development and green development. The conclusions of this paper will provide vital empirical support and reference for the role of media supervision and the realization of green and sustainable development of enterprises.

The contribution of this paper is mainly reflected in three aspects: First, existing studies mainly analyze the impact of media attention on green innovation of different types of enterprises, but less about the impact of media attention on the quality of green innovation of enterprises. This paper deeply analyzes how media attention affects enterprise green innovation and the sensitivity of media attention to different green innovation quality, and further expands the research field of the impact of media attention on enterprise innovation behavior. Secondly, the existing research mainly discusses the impact of media attention on corporate environmental behavior but does not explore the regulatory effect of environmental regulation and green financial policy between media attention and green innovation. Starting from the restraint and incentive effects of policies, this paper discusses the regulatory role of environmental regulation and green financial between media attention and green innovation and tests the heterogeneity of this regulatory effect according to the grouping of substantive and strategic green innovation, which enriches the research scope of influencing factors of enterprise green innovation.

The main structure of the present paper is as follows: the second chapter proposes the research hypotheses. The third chapter describes the sample sources and empirical models. The fourth chapter presents an empirical analysis of the impact of media attention on green innovation, analyzes the regulatory effect of environmental regulation and green finance, and carries out the robustness and endogeneity test. The fifth chapter is the discussion, which puts forward the research limitations and future research directions, and the last chapter highlights the conclusion and recommendations.

## 2. Literature Review and Research Hypotheses

### 2.1. Media Attention and Enterprise Green Innovation

Scholars have conducted plentiful research around media attention and green innovation. According to the reputation theory, media, as the primary information transmission channel between enterprises and stakeholders, serves as information intermediaries [23,24]. Negative media attention is more likely to arouse public criticism of corporate environmental responsibility than positive or favorable media attention [25], which stimulates the enthusiasm of enterprises in enhancing social reputation loss through green innovation [26]. Negative media reports on environmental pollution and excessive emissions by enterprises damages the reputation of enterprises and results in the loss of their competitive advantage in the market [27,28]. The negative media public opinion formed by media attention can make stakeholders exert pressure on enterprises [29], enhance public and other stakeholder attention to environmental protection issues [20], in turn, influencing the environmental protection behavior of enterprises, in addition to promoting green innovation among enterprises [30]. Moreover, the more negative media reports, the greater the legitimacy pressure on stakeholders, and the stronger the willingness of enterprises to innovate green products and processes quickly and effectively [21]. For instance, Kong et al. (2020) conducted research using the data of Chinese listed companies, and believed that media attention, especially negative attention, can enhance the enthusiasm of enterprises for environmental protection [31]. Wang et al. (2020) used the data of Chinese polluting enterprises for quantitative analysis and found that CEO media exposure can promote enterprises to carry out green technology innovation [32]. It is illustrated that enterprise managers, under the pressure of media attention, have strong motivation to get out of the predicament through timely and effective green innovation. Therefore, the following assumption is proposed:

**Hypothesis 1a (H1a):** *Media attention plays a positive role in promoting green innovation.*

Furthermore, media attention has prompted government agencies to pay more attention to corporate environmental performance, and enterprises may face a series of administrative penalties for environmental violations, leading to lower stock prices and downgrades of financial institutions [33,34]. In order to cope with this challenge, enterprise managers may take short-term measures to save the market reputation and meet the stakeholders' preference for "repentance" of enterprises, such as increasing the number

of strategic green innovations [21]. Nevertheless, in the long run, enterprises tend to reduce the cost of environmental compliance through substantive green innovation activities to improve the environmental performance of enterprises [35], improving the long-term profitability of enterprises to win the trust of market investors. Therefore, the following assumption is proposed:

**Hypothesis 1b (H1b):** *Media attention is conducive to promote substantive green innovation and strategic green innovation.*

### 2.2. Regulatory Effect of Environmental Regulatory Policies

As an informal environmental regulation, media attention has certain limitations [36]. Under special circumstances, the media may not have a strong direct impact on corporate behavior. In other words, the role of media attention requires other stakeholders to take action, otherwise, the influence of the media will decline or be damaged [37]. The government has a strong guiding force for the media by issuing environmental regulatory policies as a vital stakeholder [30]. Environmental regulation is an essential measure for the government to supervise enterprises to fulfill their environmental responsibilities [38,39]. Strengthening the implementation of environmental laws and regulations under the attention of the media can more effectively elevate the enthusiasm of enterprises to implement green innovation activities [40]. After the promulgation of the government's environmental regulations, the media tends to focus on discussions and reflections of the stakeholders, and the guiding effect of the implementation of the regulations on green innovation of enterprises [41], in turn, exerting external pressure on the green innovation behavior of enterprises [42,43]. Conversely, when enterprises arouse considerable social public opinion due to environmental pollution activities, government agencies will reconsider the rationality of the formulation of environmental regulations [37], or initiate a subsequent round of environmental regulation reforms [44,45], including formulation of more punitive policies to regulate enterprise environmental emissions and forcing enterprises to accelerate green innovation. Therefore, implementation of strict environmental regulation policies by governments may improve the green innovation level of enterprises under media attention. Based on this background, the following hypothesis was proposed:

**Hypothesis 2 (H2):** *Environmental regulation has a positive regulatory effect between media attention and enterprise green innovation.*

### 2.3. Regulatory Effect of Green Financial Policy

Media attention caused by environmental pollution will give rise to damage to the reputation of the enterprise, reduce the market trust, and render the enterprise to confront serious financing constraints [46,47], which makes the enterprise more motivated to rely on green transformation and development to get rid of the negative impact [31]. The government's policies supporting green transformation and development, such as the green credit policy [48,49], provide an opportunity for such enterprises. Green finance policy facilitates the acquisition of green credit funding by enterprises under the media's attention, alleviates the financing dilemma of green Research and Development (R&D) by limiting the use of funds [50], assists enterprises to recover their reputation losses through green innovation [26,31], and realizes green transformation and sustainable development by enterprises. Enterprises under negative media attention experience immense pressure to obtain loans from other financial institutions due to reputation damage and their own financial constraints [51,52]. Under the condition of green finance policy, enterprises can obtain green R&D capital by means of mortgage, alleviate their own green financial constraints, and then increase motivation for green innovation [53,54]. Therefore, the following hypothesis was proposed:

**Hypothesis 3 (H3):** *Green finance has a positive regulatory effect on media attention and green innovation.*

Based on the above analysis, the theoretical framework (Figure 1) was constructed to clarify the impact between media attention and green innovation, and the regulatory effect of environmental regulation and green finance.

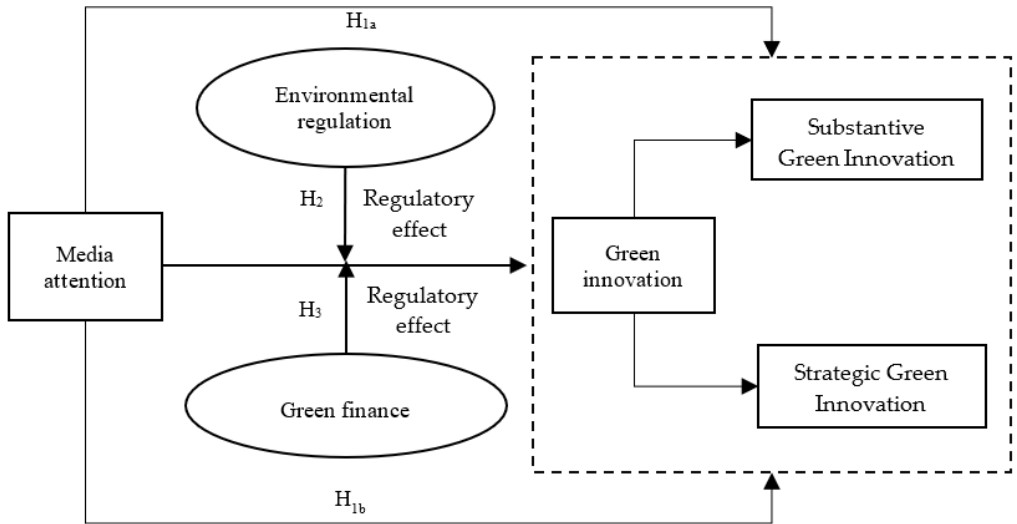

**Figure 1.** Theoretical framework of the paper.

## 3. Empirical Design

### 3.1. Data Sources

In 2010, the Chinese government promoted the integration of telecommunications networks, radio and television networks, and the Internet. At the same time, 3G, 4G, and 5G network technologies have developed rapidly since 2010, which makes the popularity of new media rise rapidly and the speed of information transmission accelerate. Consequently, we selected Chinese A-shared listed companies data from 2010 to 2019 as the research sample. The green innovation data of various enterprises were obtained from the IPC Green Inventory of China's State Patent Office and the World Intellectual Property Organization (www.wipo.int (accessed on 1 February 2020)). Media attention data were obtained from the news reports database of CSMAR (www.gtarsc.com (accessed on 1 February 2020)). The government's environmental regulation data were obtained from the *China Environmental Statistics Yearbook* and *China Statistical Yearbook*. The data used for the calculation of green finance were obtained from the *China Statistical Yearbook*, *China Insurance Yearbook*, and the *Statistical Yearbooks* of various provinces. Control variable data were obtained from the CSMAR database. To ensure the reliability of the data analyzed in the research, data for St, * ST companies and financial enterprises, and missing samples were excluded; in addition, the main continuous variables were double winsorized based on the 1% quantile. Data were analyzed using MS Excel 2016 (Microsoft Corp., Redmond, WA, USA) and Stata 16.0 (Stata Corp., College Station, TX, USA). After the above data processing, the unbalanced panel data of 9637 samples of 2970 enterprises in 9 years were finally obtained.

### 3.2. Variable Definition

#### 3.2.1. Dependent Variables

Drawing on the research of Zhao et al. [54] and Zhang et al. [55], this paper uses green invention patent and green utility model patent data to measure green innovation. The specific data acquisition process is as follows: First, the IPC number of enterprise patents issued by the China National Intellectual Property Administration is matched with the IPC number in the Green Patent Classification Inventory (Table 1) published by the World Intellectual Property Organization (WIPO) to obtain the application and authorization

amount of enterprise green invention patents and green utility models. Second, the above data is matched with the code of Chinese A-share listed companies in the CSMAR database to obtain the number of green patent applications and authorizations of Chinese enterprises.

**Table 1.** Green Patent Classification by the WIPO.

| Number | Topic | IPC (Partial Content) |
|:---:|:---|:---|
| 1 | Alternative Energy Production | Bio-fuels, Fuel cells, Harnessing energy from manmade waste, Hydro energy, etc. |
| 2 | Transportation | Vehicles in general, Vehicles other than rail vehicles, Rail vehicles, Marine vessel propulsion, etc. |
| 3 | Energy Conservation | Power supply circuitry, Low energy lighting, Thermal building insulation, in general, etc. |
| 4 | Waste Management | Treatment of waste, Reuse of waste materials, Pollution control, etc. |
| 5 | Agriculture/Forestry | Forestry techniques, Alternative irrigation techniques, Pesticide alternatives, Soil improvement, etc. |
| 6 | Administrative, Regulatory or Design Aspects | Commuting, Carbon/emissions trading, Static structure design, etc. |
| 7 | Nuclear Power Generation | Nuclear engineering, etc. |

Data source: https://www.wipo.int/classifications/ipc/green-inventory/home (accessed on 1 February 2020).

Furthermore, considering the variations between green technology patents and utility models in improving green technological innovations and economic benefits of enterprises, this paper uses the green technology patents to measure substantive green innovations and utility model patents to measure strategic green innovations [56].

3.2.2. Independent Variable

The independent variable of media attention refers to the number of times that the negative words related to the environment appear in the media reports in a certain year. Drawing on the research of Jia et al. [25] and Islam et al. [26], full texts of news reports of listed companies over the years were obtained from the CSMAR database, and the number of negative environmental media reports was manually recorded and evaluated. The specific procedures are as follows:

First, keywords that did not describe emotion and uncorrelated with environmental problems were exclude from media report keywords, such as enterprise announcements and financial reports (Irrelevant media report keywords include announcements, reports, prospectus, abstracts, latest news of listed companies, information express, corporate governance and shareholders' rights, annual reports, mid-year reports, transaction memos, full disclosure of the latest information, and overview of restructuring matters of listed companies, etc.).

Second, drawing on the research of Zhu et al. [57], the negative media report keywords associated with environmental problems (Negative media report keywords associated with environmental problems include pollution, destruction, sewage discharge, illegal discharge, wastewater, exceeding the standards, leakages, explosion, death, accident, safety, violation, smoke, oil spill, dam burst, loss, gas, carcinogens, poison, blacklist, deforestation, illegal, investigation, waste gas, residual waste, corrupt, dirty, rectification, sewage, black smoke, mildew odor, noise, random discharge, disorderly waste dumping, chromium slag, slag, manganese slag, toxic gas, mud, blood lead, waste dust, black powder, radioactivity, harmfulness, environmental violation, and environmental damage, etc.) were screened out manually from the CSMAR database.

Third, the contents of news reports based on the negative environmental report keywords associated with the media were searched and news with negative report keywords were retrieved. The news was matched with the stock code of the A-listed company based on the search results, and the number of annual media reports were counted based on the matching results, obtaining the number of negative media reports about the environment.

### 3.2.3. Adjustment Variables

(1) Environmental regulation

Environmental regulation represents the government's supervisory and managerial roles in the performance of environmental responsibilities by enterprises within its jurisdiction. It is a crucial variable to measure how the government guides enterprises to fulfill their environmental responsibilities. Based on the findings of Zhao et al. [58] and Xue et al. [59], emissions data of three main pollutants (industrial wastewater, industrial sulfur dioxide, and industrial smoke) were selected, and the entropy method was used to calculate the environmental regulation intensity index in different regions as follows:

First, the extreme value method was used to standardize pollutant emissions in different regions:

$$P_{i,j} = \frac{M_{i,j} - Min(M_{i,j})}{max(M_{i,j}) - Min(Mi_{,j})} \tag{1}$$

where $P_{i,j}$ is the standardized emission intensity of the $j$-th pollutant in region $i$, while $M_{i,j}$ is the original pollutant emissions, and $max$ $(M_{i,j})$ and $min$ $(M_{i,j})$ are the maximum and minimum values of the $j$-th pollutant in region $i$.

Second, the adjustment coefficient ($W_{i,j}$) was calculated by region based on the category of pollutants.

$$W_{i,j} = \frac{M_{i,j}}{Mean(M_{i,j})} \tag{2}$$

In formula 2, *Mean* $(M_{i,j})$ is the average value of the emission of the $j$-th pollutant in region $i$. The pollutant adjustment coefficient is used to distinguish the degree of environmental governance in different regions.

Third, the overall environmental emission intensity ($Q_i$) was calculated by regions.

$$Q_i = \frac{1}{3\sum P_{i,j} \times W_{i,j}} \tag{3}$$

Fourth, the overall environmental emission intensity ($ES_i$) was positively treated to obtain the regional environmental regulation index.

$$ES_i = \frac{Q_i - Min(Q_i)}{max(Q_i) - Min(Q_i)} \tag{4}$$

(2) Green finance

Green finance is a key government policy with regard to the use of financial instruments to stimulate green innovation among enterprises. Drawing on the research of Lee [60] and Wang [61], four indicators of green finance were selected: green investment, green credit, green insurance, and government support; in addition, the entropy method was used to calculate the green finance index. Green credit was expressed as the interest expenditure of six energy-intensive industries relative to the total local industrial interest expenditure. Green investment was expressed as the proportion of local environmental pollution control investment relative to the local GDP and green insurance was expressed as the proportion of agricultural insurance income relative to the total agricultural output value. Government support was expressed as the proportion of local environmental protection expenditure relative to the general budget expenditure. The missing values of some years were replaced by the average of the data collected over subsequent five years. Due to the lack of data from 2018 to 2019, we used the grey prediction model for prediction. The posterior error ratio C of all regional grey prediction models was 0.35. A greater GFI value indicates a relatively high development level of green finance.

### 3.2.4. Control Variables

Based on studies conducted by Wu et al. [62] and Su et al. [63], enterprise characteristic variables were selected as the control variables of the models, including financial leverage, profitability, growth ability, company value, board governance, and ownership structure. The definition of main variables is presented in Table 2.

**Table 2.** Definition of key variables.

| Variable Types | Variable Name | Variable Symbol | Variable Definition | Unit of Measurement |
|---|---|---|---|---|
| Dependent variables | Green innovation | GI | The logarithm of the total number of green patent applications after adding 1. | Piece |
| | | GA | The logarithm of the total number of green patent authorization lagging one year after adding 1. | Piece |
| | Substantive green innovation | HGI | The logarithm of the number of green invention patent applications after adding 1. | Piece |
| | | HGA | The logarithm of the number of green invention patent authorization lagging one year after adding 1. | Piece |
| | Strategic green innovation | LGI | The logarithm of the number of green utility model patent applications after adding 1. | Piece |
| | | LGA | The logarithm of the number of green utility model patent authorization lagging one year after adding 1. | Piece |
| Independent variable | Media attention | MA | The logarithm of the number of negative environmental media reports about environment. | Piece |
| Regulated variable | Environmental regulation | ER | A comprehensive ES index calculated using the entropy method for the discharge of industrial wastewater, industrial sulfur dioxide, and industrial soot in each region. | % |
| | Green finance | GF | Green investment, green credit, green insurance, and government support are selected and calculated using the entropy method. | % |
| Control variables | Financial Leverage | LEV | Asset-liability ratio, total liabilities/total assets. | % |
| | Profitability | ROA | Net interest rate on total assets, net profit/total assets. | % |
| | Growth ability | GRO | Operating income growth rate, operating income growth/total operating income. | % |
| | Company value | TQ | Tobin's Q, which equals to (equity market value + net debt market value)/total assets at the end of the period. | % |
| | Board governance | DUA | If the chairman and CEO positions are held by the same individual, the value is equal to 1, otherwise it is 0. | 1 |
| | Ownership structure | LCR | Shareholding ratio of top 10 shareholders. | % |

### 3.3. Model Construction

In order to test the necessity of including random effects in the model, the Lagrange Multiplier (LM) test of random effects was carried out. The test results showed that $p$ values are less than 0.001, indicating that the model contains random effects. In order to further determine whether the fixed effect model or the random effect model should be used, the Hausman test was conducted in this paper. The results showed that the $p$-value was 0.0000, indicating that the fixed effect model should be used. Accordingly, the following fixed effect models are constructed:

To test the impact of media attention on green innovation, model 1 was constructed as follows:

$$GIA_{it} = \alpha_0 + \alpha_1 \times MA_{it} + \alpha_2 \times LEV_{it} + \alpha_3 \times Controls_{it} + year_t + province_i + \varepsilon \quad (5)$$

To test the regulatory effect of government environmental regulation on the relationship between media attention and green innovation, model 2 was constructed as follows:

$$GIA_{it} = \beta_0 + \beta_1 \times MA_{it} + \beta_2 \times ER_{it} + \beta_3 \times MAit \times ER_{it} + \beta_4 \times Controls_{it} + year_t + province_i + \varepsilon \quad (6)$$

To test the regulatory effect of green financial policy on the relationship between media attention and green innovation, model 3 was constructed as follows:

$$GIA_{it} = \gamma_0 + \gamma_1 \times MA_{it} + \gamma_2 \times GF_{it} + \gamma_3 \times MAit \times GF_{it} + \gamma_4 \times Controls_{it} + year_t + province_i + \varepsilon \quad (7)$$

In model (1)–(3), $GIA_{it}$ represents green innovation, strategic green innovation, and substantive green innovation; $MA_{it}$ represents the number of negative media reports about environment; $year_t$ represents the time fixed effect of enterprises; $province_i$ represents the fixed effect of the province to which the enterprise belongs; $\alpha_0$, $\beta_0$, and $\gamma_0$ are the intercept terms; $\alpha_{1 \sim n}$, $\beta_{1 \sim n}$, and $\gamma_{1 \sim n}$ are the correlation coefficients; and $\varepsilon$ represents the random disturbance terms.

Since the data used in this paper are non-equilibrium and belong to the short panel, in turn, the time dimension T is small, which indicates that the information of individual enterprises is relatively few. Accordingly, it is impossible to find out whether there is autocorrelation in the disturbance term, which is generally assumed independent and identically distributed, so the test of serial correlation can be selected not to be carried out.

In order to verify whether there is heteroscedasticity in the data, the White test is used in this paper. The results show that the $p$-value is less than 0.001, which strongly rejects the original hypothesis, indicating that the data have a heteroscedasticity problem. In order to solve the heteroscedasticity problem, this paper takes the logarithm of the key variables to eliminate the influence of heteroscedasticity.

In addition, the method of the Pesaran test is used to verify the cross-sectional correlation of the data. The results show that the $p$-value is less than 0.01, illustrating that there is cross-sectional correlation. Referring to Wooldridge (2002) [64] and Petersen (2009) [65], the method of robust standard errors clustered to individual firms is adopted to correct the autocorrelation and heteroscedasticity problems. On the one hand, this method is not sensitive to the heteroscedasticity and autocorrelation problems that may exist in the model, so it can improve the accuracy of estimation. On the other hand, because the random disturbance term of the same individual in different years has correlation, the use of robust standard error can better capture the characteristics of correlation within the group, so as to obtain a consistent estimate of the true standard error.

### 3.4. Descriptive Statistics

The descriptive statistical analyzed results of the selected variables are presented in Table 3. The maximum and minimum values for the total number of green patent applications of sampled enterprises are 7.0851 and 0, and the average value is 0.479. In contrast, the maximum and minimum values for the total number of green patent

authorizations are 6.941 and 0, and the average value is 0.836, which indicates that the levels of green innovation of Chinese listed companies were significantly different. The maximum and minimum values of media attention are 5.565 and 0.693, respectively, with an average of 1.093, suggesting that the media's attention to the negative environmental problems of enterprises is uneven. The maximum and minimum values of green finance are 0.793 and 0.06, suggesting that green finance policies among the regions varied considerably. Pearson's correlation test was carried out to determine the correlations among key variables. The average variance inflation factor between variables was 1.39, which eliminated the problem of high multicollinearity.

**Table 3.** Descriptive statistics.

| Variables | Observation | Mean | S.D. | Min | Max |
|---|---|---|---|---|---|
| GI | 9637 | 0.479 | 0.96 | 0 | 7.085 |
| HGI | 9637 | 0.341 | 0.801 | 0 | 6.59 |
| LGI | 9637 | 0.289 | 0.706 | 0 | 6.146 |
| GA | 9637 | 0.836 | 1.173 | 0 | 6.941 |
| HGA | 9637 | 0.363 | 0.782 | 0 | 6.282 |
| LGA | 9637 | 0.694 | 1.074 | 0 | 6.537 |
| ES | 9637 | 0.646 | 0.306 | 0 | 1 |
| GF | 9637 | 0.263 | 0.156 | 0.06 | 0.793 |
| MA | 9637 | 1.093 | 0.565 | 0.693 | 5.565 |
| LEV | 9637 | 0.429 | 0.216 | 0.049 | 0.98 |
| GRO | 9345 | 0.423 | 1.204 | −0.789 | 10.288 |
| TQ | 9486 | 1.042 | 0.341 | 0.537 | 5.561 |
| ROA | 9637 | 0.042 | 0.072 | 0-.366 | 0.206 |
| DUA | 9525 | 0.28 | 0.449 | 0 | 1 |
| LCR | 9637 | 4.07 | 0.297 | 1.523 | 4.601 |

## 4. Empirical Results

### 4.1. Impact of Media Attention on Enterprise Green Innovation

The empirical results of the impact of media attention on enterprise green innovation are presented in Table 4. It can be seen from columns (1) and (2) that the influence coefficient of media attention on the total number of green innovation applications was 0.374, and the influence coefficient of media attention on the total number of green invention authorization was 0.395, both of which are significant at the level of 1%, which indicates that media attention is conductive to improve the green innovation. Hypothesis 1a has been verified.

**Table 4.** Empirical results of the impact of media attention on green innovation.

| Variables | (1) | (2) | (3) | (4) | (5) | (6) |
|---|---|---|---|---|---|---|
| | GI | GA | HGI | HGA | LGI | LGA |
| MA | 0.374 *** | 0.395 *** | 0.342 *** | 0.316 *** | 0.244 *** | 0.326 *** |
| | (0.0293) | (0.0282) | (0.0267) | (0.0229) | (0.0231) | (0.0263) |
| LEV | 0.360 *** | 1.066 *** | 0.297 *** | 0.555 *** | 0.300 *** | 1.016 *** |
| | (0.0483) | (0.0601) | (0.0398) | (0.0395) | (0.0357) | (0.0556) |
| GRO | −0.0377 *** | −0.0580 *** | −0.0250 *** | −0.0251 *** | −0.0315 *** | −0.0542 *** |
| | (0.00491) | (0.00788) | (0.00410) | (0.00486) | (0.00337) | (0.00709) |
| TQ | −0.282 *** | −0.569 *** | −0.207 *** | −0.312 *** | −0.225 *** | −0.499 *** |
| | (0.0305) | (0.0356) | (0.0257) | (0.0232) | (0.0221) | (0.0329) |
| ROA | 0.898 *** | 1.551 *** | 0.710 *** | 0.711 *** | 0.575 *** | 1.421 *** |
| | (0.133) | (0.162) | (0.110) | (0.100) | (0.0974) | (0.149) |
| DUA | 0.0297 | −0.0557 * | 0.0195 | −0.0414 * | 0.0252 | −0.0368 |
| | (0.0224) | (0.0251) | (0.0188) | (0.0171) | (0.0170) | (0.0232) |

**Table 4.** *Cont.*

| Variables | (1) | (2) | (3) | (4) | (5) | (6) |
|---|---|---|---|---|---|---|
| | GI | GA | HGI | HGA | LGI | LGA |
| LCR | −0.0815 * | −0.0974 * | −0.0946 *** | −0.0771 ** | 0.0151 | −0.0196 |
| | (0.0332) | (0.0394) | (0.0285) | (0.0278) | (0.0237) | (0.0359) |
| α | 0.354 * | 0.925 *** | 0.320 * | 0.537 *** | −0.0863 | 0.468 ** |
| | (0.156) | (0.184) | (0.135) | (0.129) | (0.109) | (0.166) |
| Time | Yes | Yes | Yes | Yes | Yes | Yes |
| Province | Yes | Yes | Yes | Yes | Yes | Yes |
| N | 9091 | 9091 | 9091 | 9091 | 9091 | 9091 |
| $r^2$ | 0.102 | 0.188 | 0.106 | 0.153 | 0.085 | 0.173 |
| F | 16.25 | 34.35 | 14.10 | 19.52 | 12.64 | 30.32 |
| Hausman_chi2 | 253.73 | 332.46 | 300.09 | 393.04 | 238.61 | 264.40 |

Standard errors in parentheses; *** $p < 0.01$, ** $p < 0.05$, * $p < 0.1$.

The empirical results of the impact of media attention on substantive green innovation and strategic green innovation of enterprises are presented in columns (3)–(6). The impact coefficients of media attention on green innovation application and authorization are 0.342 and 0.316, respectively, which are significant at the 1% level, and the impact coefficients of media attention on strategic green innovation are 0.244 and 0.326, which are significant at the 1% level. The results indicate that media attention has a positive impact on both substantive and strategic green innovation, which verified Hypothesis 1b.

### 4.2. Regulatory Effect of Environmental Regulation

The regulatory effect of environmental regulation on the interaction between media attention and enterprise green innovation is presented in Table 5. It can be seen from columns (1) and (2) that the impact coefficient of the intersection of environmental regulation and media attention (ES × MA) on green innovation applications is 0.206, which is significant at the 10% level, while on green innovation authorization, it is 0.334, which is significant at the 1% level. The results reveal that environmental regulation has a positive regulatory effect on the relationship between media attention and enterprise green innovation, which verified Hypothesis 2.

**Table 5.** Empirical results of the regulatory effect of environmental regulation.

| Variables | (1) | (2) | (3) | (4) | (5) | (6) |
|---|---|---|---|---|---|---|
| | Model 2 | | | | | |
| | GI | GA | HGI | HGA | LGI | LGA |
| MA | 0.372 *** | 0.392 *** | 0.340 *** | 0.314 *** | 0.243 *** | 0.323 *** |
| | (0.0291) | (0.0279) | (0.0265) | (0.0228) | (0.0229) | (0.0260) |
| ES | −0.0279 | 0.218 | −0.0793 | 0.0691 | 0.0752 | 0.190 |
| | (0.134) | (0.152) | (0.113) | (0.102) | (0.0984) | (0.140) |
| ES × MA | 0.206 * | 0.334 *** | 0.178 * | 0.254 *** | 0.154 * | 0.310 *** |
| | (0.0850) | (0.0839) | (0.0771) | (0.0671) | (0.0649) | (0.0785) |
| LEV | 0.362 *** | 1.070 *** | 0.299 *** | 0.558 *** | 0.302 *** | 1.020 *** |
| | (0.0482) | (0.0600) | (0.0397) | (0.0394) | (0.0357) | (0.0555) |
| ROA | 0.916 *** | 1.572 *** | 0.727 *** | 0.729 *** | 0.585 *** | 1.440 *** |
| | (0.134) | (0.162) | (0.111) | (0.101) | (0.0975) | (0.149) |
| GRO | −0.0371 *** | −0.0570 *** | −0.0245 *** | −0.0244 *** | −0.0311 *** | −0.0533 *** |
| | (0.00489) | (0.00788) | (0.00407) | (0.00484) | (0.00337) | (0.00709) |
| TQ | −0.279 *** | −0.561 *** | −0.205 *** | −0.307 *** | −0.222 *** | −0.492 *** |
| | (0.0304) | (0.0357) | (0.0256) | (0.0232) | (0.0220) | (0.0329) |
| DUA | 0.0299 | −0.0552 * | 0.0198 | −0.0411 * | 0.0254 | −0.0363 |
| | (0.0225) | (0.0251) | (0.0188) | (0.0171) | (0.0170) | (0.0232) |

**Table 5.** *Cont.*

| Variables | (1) | (2) | (3) | (4) | (5) | (6) |
|---|---|---|---|---|---|---|
| | | | **Model 2** | | | |
| | **GI** | **GA** | **HGI** | **HGA** | **LGI** | **LGA** |
| LCR | −0.0857 ** | −0.104 ** | −0.0982 *** | −0.0822 ** | 0.0120 | −0.0257 |
| | (0.0331) | (0.0393) | (0.0284) | (0.0277) | (0.0236) | (0.0359) |
| α | 0.386 | 0.731 ** | 0.400 * | 0.480 ** | −0.152 | 0.300 |
| | (0.203) | (0.230) | (0.173) | (0.158) | (0.144) | (0.210) |
| Time | Yes | Yes | Yes | Yes | Yes | Yes |
| Province | Yes | Yes | Yes | Yes | Yes | Yes |
| *N* | 9091 | 9091 | 9091 | 9091 | 9091 | 9091 |
| r$^2$ | 0.103 | 0.190 | 0.108 | 0.156 | 0.086 | 0.175 |
| F | 15.52 | 33.40 | 13.48 | 18.96 | 12.15 | 29.54 |
| Hausman_chi2 | 258.63 | 338.45 | 304.36 | 405.20 | 244.43 | 272.06 |

Standard errors in parentheses; *** $p < 0.01$, ** $p < 0.05$, * $p < 0.1$.

In order to further analyze the differences in the regulatory effects of environmental regulation, the strategic green innovation group and substantive green innovation group were divided to test the heterogeneity respectively. As shown in columns (3) and (4), the impact coefficient of the intersection of environmental regulation and media attention (ES × MA) on substantive green innovation applications and authorizations are 0.178 and 0.254, which are significant at the 10% and 1% level, while the impact coefficient on strategic green innovation applications and authorizations are 0.154 and 0.310, respectively, which are significant at the 10% and 1% level. This demonstrates that the regulatory effect of environmental regulation is positive in improving substantive and strategic green innovation for enterprises confronted with media pressure, which verified Hypothesis 2 again.

*4.3. Regulatory Effect of Green Finance*

The regulatory effect of green finance on the impact of media attention and green innovation is presented in Table 6. It can be seen from columns (1) and (2) that the impact coefficient of the intersection of green finance and media attention (GF × MA) on green innovation applications is 0.206, and the impact coefficient on green innovation authorizations is 0.745, which are significant at the 10% and 1% level. The results suggest that green finance has a significant positive regulatory effect on the impact of media attention on enterprise green innovation, and this regulatory effect is more significant on the effective green innovation, which verified Hypothesis 3.

**Table 6.** Empirical results of the regulatory effect of green finance.

| Variables | (1) | (2) | (3) | (4) | (5) | (6) |
|---|---|---|---|---|---|---|
| | | | **Model 3** | | | |
| | **GI** | **GA** | **HGI** | **HGA** | **LGI** | **LGA** |
| MA | 0.372 *** | 0.381 *** | 0.340 *** | 0.300 *** | 0.243 *** | 0.315 *** |
| | (0.0291) | (0.0270) | (0.0265) | (0.0212) | (0.0229) | (0.0253) |
| GF | −0.0279 | 0.0525 | −0.0793 | 0.631 * | 0.0752 | −0.151 |
| | (0.134) | (0.363) | (0.113) | (0.268) | (0.0984) | (0.341) |
| GF × MA | 0.206 * | 0.745 *** | 0.178 * | 0.908 *** | 0.154 * | 0.522 ** |
| | (0.0850) | (0.193) | (0.0771) | (0.161) | (0.0649) | (0.183) |
| LEV | 0.362 *** | 1.066 *** | 0.299 *** | 0.551 *** | 0.302 *** | 1.017 *** |
| | (0.0482) | (0.0600) | (0.0397) | (0.0393) | (0.0357) | (0.0556) |
| GRO | −0.0371 *** | −0.0563 *** | −0.0245 *** | −0.0230 *** | −0.0311 *** | −0.0531 *** |
| | (0.00489) | (0.00787) | (0.00407) | (0.00483) | (0.00337) | (0.00708) |

**Table 6.** *Cont.*

| Variables | (1) | (2) | (3) | (4) | (5) | (6) |
|---|---|---|---|---|---|---|
| | GI | GA | HGI | HGA | LGI | LGA |
| | | | Model 3 | | | |
| TQ | −0.279 *** | −0.556 *** | −0.205 *** | −0.298 *** | −0.222 *** | −0.489 *** |
| | (0.0304) | (0.0357) | (0.0256) | (0.0230) | (0.0220) | (0.0330) |
| ROA | 0.916 *** | 1.606 *** | 0.727 *** | 0.779 *** | 0.585 *** | 1.459 *** |
| | (0.134) | (0.163) | (0.111) | (0.103) | (0.0975) | (0.150) |
| DUA | 0.0299 | −0.0536 * | 0.0198 | −0.0387 * | 0.0254 | −0.0354 |
| | (0.0225) | (0.0251) | (0.0188) | (0.0170) | (0.0170) | (0.0232) |
| LCR | −0.0857 ** | −0.0964 * | −0.0982 *** | −0.0714 * | 0.0120 | −0.0203 |
| | (0.0331) | (0.0395) | (0.0284) | (0.0277) | (0.0236) | (0.0360) |
| α | 0.386 | 0.879 ** | 0.400 * | 0.163 | −0.152 | 0.542 * |
| | (0.203) | (0.273) | (0.173) | (0.195) | (0.144) | (0.253) |
| Time | Yes | Yes | Yes | Yes | Yes | Yes |
| Province | Yes | Yes | Yes | Yes | Yes | Yes |
| *N* | 9091 | 9091 | 9091 | 9091 | 9091 | 9091 |
| $r^2$ | 0.103 | 0.191 | 0.108 | 0.165 | 0.086 | 0.174 |
| F | 16.32 | 34.06 | 14.44 | 19.86 | 12.45 | 30.10 |
| Hausman_chi2 | 257.26 | 341.88 | 306.32 | 417.52 | 241.67 | 273.14 |

Standard errors in parentheses; *** $p < 0.01$, ** $p < 0.05$, * $p < 0.1$.

As presented in columns (4) and (6), the impact coefficient of the intersection of green finance and media attention on substantive green innovation is 0.908, which is significant at the 1% level, while the impact coefficient of the intersection on strategic green innovation is 0.522, which is significant at the 5% level. This illustrates that the implementation of green finance policy is more effective in improving the substantive green innovation when under the pressure of negative media attention due to environmental problems.

### 4.4. Heterogeneity Analysis

In order to further test the heterogeneous impact of media attention on diverse green innovation, and the difference in the regulatory effect of environmental regulation and green finance, so as to provide targeted recommendations for stimulating enterprise green innovation, this paper conducts empirical tests grouped by whether belonging to heavy polluting enterprises and state-owned enterprises.

#### 4.4.1. The Heterogeneous Influence of Media Attention on Green Innovation

The empirical test results of the heterogeneous impact of media attention on green innovation are presented in Table 7. Columns (1) and (2) respectively exhibit the empirical results of the impact of media attention on the green innovation of heavy polluting enterprises and non-heavy polluting enterprises. It can be seen that the impact coefficients of media attention on the green innovation of heavy polluting enterprises and non-heavy polluting enterprises are 0.417 and 0.338, respectively, both of which are significant at the 1% level, but the impact coefficient of heavy polluting enterprises is greater than that of non-heavy polluting enterprises, which indicates that media attention is more conducive to motivating the green innovation output of heavy polluting enterprises.

**Table 7.** Empirical results of heterogeneous influence of media attention on green innovation.

| Variables | (1) Heavy Pollution Group | (2) Non-Heavy Pollution Group | (3) State-Owned Enterprise Group | (4) Non-State Owned Enterprise Group |
|---|---|---|---|---|
| | GIAS | GIAS | GIAS | GIAS |
| MA | 0.417 *** | 0.338 *** | 0.466 *** | 0.268 *** |
| | (0.0513) | (0.0332) | (0.0477) | (0.0364) |
| LEV | 0.0240 | 0.443 *** | −0.0116 | 0.502 *** |
| | (0.0846) | (0.0573) | (0.102) | (0.0609) |
| GRO | −0.0401 *** | −0.0405 *** | −0.0431 *** | −0.0284 *** |
| | (0.0101) | (0.00560) | (0.00903) | (0.00549) |
| TQ | −0.453 *** | −0.249 *** | −0.380 *** | −0.193 *** |
| | (0.0554) | (0.0354) | (0.0637) | (0.0330) |
| ROA | 0.0156 | 1.138 *** | 0.291 | 0.969 *** |
| | (0.260) | (0.155) | (0.395) | (0.132) |
| DUA | −0.0955 ** | 0.0712 ** | 0.0940 | 0.0168 |
| | (0.0340) | (0.0271) | (0.0647) | (0.0242) |
| LCR | 0.0706 | −0.168 *** | −0.117 | 0.0108 |
| | (0.0480) | (0.0408) | (0.0637) | (0.0381) |
| Cons | 0.453 * | 0.584 ** | 0.653 * | −0.0121 |
| | (0.230) | (0.194) | (0.298) | (0.181) |
| Time | Yes | Yes | Yes | Yes |
| Province | Yes | Yes | Yes | Yes |
| $N$ | 2379 | 6712 | 3371 | 5548 |
| $r^2$ | 0.174 | 0.102 | 0.128 | 0.091 |
| F | 6.77 | 15.51 | 7.18 | 14.25 |
| Hausman_chi2 | 115.47 | 162.43 | 112.03 | 176.16 |

Standard errors in parentheses; *** $p < 0.01$, ** $p < 0.05$, * $p < 0.1$.

Columns (3) and (4) are the empirical results of the impact of media attention on green innovation of state-owned enterprises and non-state-owned enterprises. It manifests that the impact coefficients of media attention on the green innovation of state-owned enterprises and non-state-owned enterprises are significant at the level of 1%, but the impact coefficient of state-owned enterprises is greater than that of non-state-owned enterprises, which demonstrates that green innovation is more conducive to stimulating green innovation output of state-owned enterprises.

4.4.2. Heterogeneity of Regulatory Effect of Environmental Regulation

Table 8 shows the empirical test results of the heterogeneity of the regulatory effect of environmental regulation. Columns (1) and (2) present the empirical results of the regulatory effect of environmental regulation on heavy polluting enterprises and non-heavy polluting enterprises, respectively. The influence coefficient of the interaction item of environmental regulation and media attention in the heavy pollution group is 0.377, which is significant at the level of 1%, but the interaction item coefficient of heavy pollution enterprises is not significant, which indicates that under the pressure from public opinion, environmental regulation is more conducive to boosting the green innovation of heavy pollution enterprises. By comparing columns (3) and (4), the influence coefficient of interaction between media attention and green innovation in the state-owned group is 0.379, which is significant at the level of 5%, while the influence coefficient in the non-state-owned enterprise group is not significant, which indicates that environmental regulation plays a more active role in raising the green innovation of state-owned enterprises.

**Table 8.** Empirical results of heterogeneity of the regulatory effect of environmental regulation.

| Variables | (1) Heavy Pollution Group | (2) Non-Heavy Pollution Group | (3) State-Owned Enterprise Group | (4) Non-State-Owned Enterprise Group |
|---|---|---|---|---|
| | GIAS | GIAS | GIAS | GIAS |
| MA | 0.377 *** | 0.338 *** | 0.430 *** | 0.267 *** |
| | (0.0427) | (0.0332) | (0.0484) | (0.0359) |
| ES | 0.0203 | −0.0503 | −0.530 * | 0.236 |
| | (0.192) | (0.174) | (0.226) | (0.168) |
| ES × MA | 0.864 *** | −0.0198 | 0.379 ** | −0.0267 |
| | (0.129) | (0.0957) | (0.147) | (0.0931) |
| LEV | 0.0608 | 0.442 *** | 0.00927 | 0.503 *** |
| | (0.0830) | (0.0573) | (0.100) | (0.0610) |
| GRO | −0.0399 *** | −0.0405 *** | −0.0431 *** | −0.0284 *** |
| | (0.00976) | (0.00560) | (0.00908) | (0.00550) |
| TQ | −0.434 *** | −0.250 *** | −0.364 *** | −0.191 *** |
| | (0.0550) | (0.0354) | (0.0626) | (0.0329) |
| ROA | 0.113 | 1.137 *** | 0.347 | 0.962 *** |
| | (0.260) | (0.155) | (0.392) | (0.133) |
| DUA | −0.0829 * | 0.0713 ** | 0.103 | 0.0168 |
| | (0.0337) | (0.0271) | (0.0647) | (0.0242) |
| LCR | 0.0700 | −0.167 *** | −0.127 * | 0.0105 |
| | (0.0484) | (0.0408) | (0.0631) | (0.0381) |
| Cons | 0.360 | 0.630 * | 1.186 ** | −0.223 |
| | (0.303) | (0.256) | (0.365) | (0.243) |
| Time | Yes | Yes | Yes | Yes |
| Province | Yes | Yes | Yes | Yes |
| *N* | 2379 | 6712 | 3371 | 5548 |
| r$^2$ | 0.203 | 0.102 | 0.133 | 0.091 |
| F | 7.06 | 15.08 | 7.06 | 13.79 |
| Hausman_chi2 | 150.57 | 164.89 | 122.77 | 181.49 |

Standard errors in parentheses; *** $p < 0.01$, ** $p < 0.05$, * $p < 0.1$.

### 4.4.3. Heterogeneity of the Regulation Effect of Green Finance

Table 9 exhibits the empirical test results of the heterogeneity of the regulatory effect of environmental regulation. The result of columns (1) and (2) manifests that the influence coefficient of the interaction item of green finance and media attention in the heavy pollution group is 2.261, which is significant at the level of 1%. Nevertheless, the influence coefficient of the interaction item in the heavy pollution group is not significant, which indicates that under the pressure of the media, green finance is more helpful in improving the green innovation output of heavy pollution enterprises. Corresponding, by comparing columns (3) and (4), it is found that the interaction coefficient between media attention and green innovation in the state-owned group is 1.047, which is significant at the level of 1%, while the interaction coefficient of the non-state-owned group is not significant, which indicates that green finance is more effective in improving the green innovation output of state-owned enterprises.

**Table 9.** Empirical results of heterogeneity of the regulatory effect of green finance.

| Variables | (1) Heavy Pollution Group | (2) Non-Heavy Pollution Group | (3) State-Owned Enterprise Group | (4) Non-State Owned Enterprise Group |
|---|---|---|---|---|
| | GIAS | GIAS | GIAS | GIAS |
| MA | 0.424 *** | 0.342 *** | 0.440 *** | 0.268 *** |
| | (0.0387) | (0.0330) | (0.0429) | (0.0363) |

**Table 9.** *Cont.*

| Variables | (1) Heavy Pollution Group | (2) Non-Heavy Pollution Group | (3) State-Owned Enterprise Group | (4) Non-State Owned Enterprise Group |
|---|---|---|---|---|
| | GIAS | GIAS | GIAS | GIAS |
| GF | 0.992 | −0.0205 | 0.176 | 0.116 |
| | (0.715) | (0.308) | (0.486) | (0.356) |
| GF × MA | 2.261 *** | −0.169 | 1.047 *** | 0.0575 |
| | (0.265) | (0.163) | (0.264) | (0.197) |
| LEV | 0.0730 | 0.443 *** | 0.00488 | 0.501 *** |
| | (0.0814) | (0.0573) | (0.100) | (0.0609) |
| GRO | −0.0390 *** | −0.0408 *** | −0.0390 *** | −0.0283 *** |
| | (0.00921) | (0.00561) | (0.00899) | (0.00549) |
| TQ | −0.357 *** | −0.250 *** | −0.340 *** | −0.193 *** |
| | (0.0514) | (0.0355) | (0.0628) | (0.0330) |
| ROA | 0.267 | 1.128 *** | 0.385 | 0.973 *** |
| | (0.248) | (0.155) | (0.392) | (0.132) |
| DUA | −0.0986 ** | 0.0707 ** | 0.0982 | 0.0168 |
| | (0.0335) | (0.0271) | (0.0640) | (0.0241) |
| LCR | 0.0680 | −0.169 *** | −0.119 | 0.0125 |
| | (0.0478) | (0.0408) | (0.0632) | (0.0384) |
| Cons | −0.375 | 0.596 * | 0.508 | −0.0815 |
| | (0.430) | (0.254) | (0.366) | (0.272) |
| Time | Yes | Yes | Yes | Yes |
| Province | Yes | Yes | Yes | Yes |
| N | 2379 | 6712 | 3371 | 5548 |
| r² | 0.238 | 0.102 | 0.140 | 0.091 |
| F | 9.48 | 14.87 | 7.49 | 13.77 |
| Hausman_chi2 | 157.06 | 167.05 | 117.67 | 177.90 |

Standard errors in parentheses; *** $p < 0.01$, ** $p < 0.05$, * $p < 0.1$.

*4.5. Robustness Test*

The above research proves that media attention has a significant role in promoting green innovation of enterprises, and that environmental regulation and green finance play a positive regulatory effect. In order to verify the robustness of these empirical results, this paper uses the lag data of green innovation applications as a surrogate variable to conduct the further tests.

4.5.1. Robustness Test of Impact of Media Attention on Enterprise Green Innovation

It should be noted from Table 10 that the impact of media attention on enterprise green innovation is still significantly positive after using the lag data of green innovation applications as a surrogate variable, indicating that media attention promotes enterprise green innovation again. The results of columns (2) and (3) present that media attention is conducive to boosting both substantive and strategic green innovation, which is consistent with the above results, indicating that the conclusions of assumption 1a and 1b are robust.

**Table 10.** Robustness test results of the impact of media attention on green innovation.

| Variables | (1) | (2) | (3) |
|---|---|---|---|
| | | Model 1 | |
| | GI-1 | HGI-1 | LGI-1 |
| MC | 0.364 *** | 0.330 *** | 0.237 *** |
| | (0.0289) | (0.0263) | (0.0227) |
| LEV | 0.389 *** | 0.322 *** | 0.294 *** |
| | (0.0478) | (0.0402) | (0.0344) |

**Table 10.** *Cont.*

| Variables | (1) | (2) | (3) |
|---|---|---|---|
| | **Model 1** | | |
| | **GI-1** | **HGI-1** | **LGI-1** |
| GRO | −0.0334 *** | −0.0227 *** | −0.0275 *** |
| | (0.00540) | (0.00462) | (0.00347) |
| TQ | −0.260 *** | −0.200 *** | −0.203 *** |
| | (0.0298) | (0.0262) | (0.0202) |
| ROA | 1.183 *** | 0.976 *** | 0.687 *** |
| | (0.123) | (0.102) | (0.0904) |
| DUA | 0.0306 | 0.0135 | 0.0353 * |
| | (0.0221) | (0.0188) | (0.0164) |
| LCR | −0.0498 | −0.0793 ** | 0.0390 |
| | (0.0332) | (0.0289) | (0.0229) |
| $\alpha$ | 0.248 | 0.293 * | −0.170 |
| | (0.155) | (0.136) | (0.104) |
| Time | Yes | Yes | Yes |
| Province | Yes | Yes | Yes |
| $N$ | 9077 | 9077 | 9077 |
| $r^2$ | 0.100 | 0.102 | 0.087 |
| F | 14.81 | 12.99 | 12.18 |
| Hausman_chi2 | 200.43 | 235.87 | 207.79 |

Standard errors in parentheses; *** $p < 0.01$, ** $p < 0.05$, * $p < 0.1$.

#### 4.5.2. Robustness Test of the Regulatory Effect of Environmental Regulation and Green Finance

The robustness test results of the regulatory effects of environmental regulation and green finance are shown in Table 11. The results of columns (1) and (4) illustrate that the regulatory effect of environmental regulation and green finance between media attention and green innovation is still significantly positive after using the lag data of green innovation applications as a surrogate variable. By comparing the results of columns (2) and (3), it is verified that the regulatory effect of environmental regulation is significant both in the substantive green innovation and strategic green innovation. In addition, it is found from columns (5) and (6) that the regulatory effect of green finance is more significant in the substantive green innovation, which is consistent with the above conclusions, indicating that the conclusions of assumptions are robust.

**Table 11.** Robustness test results of regulatory effects of environmental regulation and green finance.

| Variables | (1) | (2) | (3) | (4) | (5) | (6) |
|---|---|---|---|---|---|---|
| | **Model 2** | | | **Model 3** | | |
| | **GI-1** | **HGI-1** | **LGI-1** | **GI-1** | **HGI-1** | **LGI-1** |
| MC | 0.363 *** | 0.328 *** | 0.236 *** | 0.357 *** | 0.321 *** | 0.231 *** |
| | (0.0287) | (0.0261) | (0.0225) | (0.0276) | (0.0249) | (0.0216) |
| ES | 0.0913 | −0.00776 | 0.121 | | | |
| | (0.136) | (0.113) | (0.103) | | | |
| GF | | | | 0.0829 | 0.0977 | −0.0140 |
| | | | | (0.281) | (0.249) | (0.193) |
| ES × MC | 0.216 ** | 0.190 * | 0.137 * | | | |
| | (0.0837) | (0.0754) | (0.0646) | | | |
| GF × MC | | | | 0.434 * | 0.496 ** | 0.298 * |
| | | | | (0.176) | (0.162) | (0.136) |
| LEV | 0.391 *** | 0.324 *** | 0.296 *** | 0.389 *** | 0.322 *** | 0.294 *** |
| | (0.0477) | (0.0401) | (0.0343) | (0.0478) | (0.0402) | (0.0344) |

**Table 11.** *Cont.*

| Variables | (1) | (2) | (3) | (4) | (5) | (6) |
|---|---|---|---|---|---|---|
| | Model 2 | | | Model 3 | | |
| | GI-1 | HGI-1 | LGI-1 | GI-1 | HGI-1 | LGI-1 |
| GRO | −0.0328 *** | −0.0221 *** | −0.0271 *** | −0.0325 *** | −0.0216 *** | −0.0269 *** |
| | (0.00536) | (0.00458) | (0.00345) | (0.00539) | (0.00460) | (0.00346) |
| TQ | −0.255 *** | −0.197 *** | −0.200 *** | −0.252 *** | −0.191 *** | −0.197 *** |
| | (0.0298) | (0.0262) | (0.0201) | (0.0297) | (0.0260) | (0.0200) |
| ROA | 1.197 *** | 0.991 *** | 0.694 *** | 1.214 *** | 1.011 *** | 0.708 *** |
| | (0.123) | (0.102) | (0.0904) | (0.123) | (0.102) | (0.0905) |
| DUA | 0.0309 | 0.0138 | 0.0356 * | 0.0319 | 0.0151 | 0.0362 * |
| | (0.0221) | (0.0188) | (0.0164) | (0.0221) | (0.0187) | (0.0164) |
| LCR | −0.0541 | −0.0831 ** | 0.0363 | −0.0489 | −0.0782 ** | 0.0391 |
| | (0.0330) | (0.0288) | (0.0227) | (0.0331) | (0.0289) | (0.0229) |
| $\alpha$ | 0.169 | 0.306 | −0.279 | 0.192 | 0.228 | −0.169 |
| | (0.202) | (0.173) | (0.143) | (0.218) | (0.192) | (0.151) |
| Time | Yes | Yes | Yes | Yes | Yes | Yes |
| Province | Yes | Yes | Yes | Yes | Yes | Yes |
| N | 9077 | 9077 | 9077 | 9077 | 9077 | 9077 |
| $r^2$ | 0.101 | 0.103 | 0.088 | 0.101 | 0.105 | 0.089 |
| F | 14.15 | 12.44 | 11.68 | 14.70 | 13.01 | 11.91 |
| Hausman_chi2 | 202.53 | 238.87 | 210.42 | 205.55 | 245.41 | 211.21 |

Standard errors in parentheses; *** $p < 0.01$, ** $p < 0.05$, * $p < 0.1$.

### 4.6. Endogenous Test

In order to solve the endogenous problem in the empirical test, this paper uses the instrumental variable method and the two-stage least square method to execute further tests. Referring to Zheng and Ward (2011) [66] and Li et al. (2022) [19], this paper uses the average number of industry media reports about the environment in the current year as the instrumental variable. It can be seen from Table 12 that the Kleibergen–Paap rk LM statistic is significant at the level of 1%, and the original hypothesis that the selected instrumental variable is not sufficiently identified should be rejected; Cragg Donald Wald F statistics are all greater than the critical value of the Stock–Yogo weak instrumental variable identification F-test at the 10% significance level. The original hypothesis of the weak instrumental variable should be rejected, which indicates that the selected instrumental variable is reasonable. The empirical results in Table 12 demonstrate that the impact coefficient of media attention on green innovation in the second stage is still significantly positive after the instrumental variables are added, indicating that the main conclusions of this paper are still consistent.

**Table 12.** Results of the endogenous test.

| Variables | First Stage | Second Stage | First Stage | Second Stage | First Stage | Second Stage |
|---|---|---|---|---|---|---|
| | Model 1 | | Model 2 | | Model 3 | |
| | MA | GI | MA | GI | MA | GI |
| IV | 1.065 *** | | 1.061 *** | | 1.048 *** | |
| | (0.082) | | (0.081) | | (0.079) | |
| MA | | 0.497 *** | | 0.487 *** | | 0.475 *** |
| | | (0.093) | | (0.091) | | (0.092) |
| ES | | | 0.002 | −0.024 | | |
| | | | (0.078) | (0.134) | | |
| GF | | | | | −0.465 *** | 0.179 |
| | | | | | (0.173) | (0.291) |

**Table 12.** *Cont.*

| Variables | First Stage | Second Stage | First Stage | Second Stage | First Stage | Second Stage |
|---|---|---|---|---|---|---|
| | Model 1 | | Model 2 | | Model 3 | |
| | **MA** | **GI** | **MA** | **GI** | **MA** | **GI** |
| MA × ES | | | 0.079 | 0.195 ** | | |
| | | | (0.087) | (0.084) | | |
| MA × FG | | | | | 0.563 *** | 0.477 *** |
| | | | | | (0.150) | (0.185) |
| LEV | 0.433 *** | 0.313 *** | 0.433 *** | 0.318 *** | 0.431 *** | 0.317 *** |
| | (0.031) | (0.061) | (0.031) | (0.061) | (0.031) | (0.061) |
| GRO | −0.011 ** | −0.037 *** | −0.010 ** | −0.036 *** | −0.009 ** | −0.036 *** |
| | (0.004) | (0.005) | (0.004) | (0.005) | (0.004) | (0.005) |
| TQ | 0.117 *** | −0.299 *** | 0.118 *** | −0.294 *** | 0.127 *** | −0.289 *** |
| | (0.021) | (0.033) | (0.021) | (0.033) | (0.021) | (0.033) |
| ROA | 0.800 *** | 0.807 *** | 0.806 *** | 0.830 *** | 0.831 *** | 0.853 *** |
| | (0.099) | (0.153) | (0.098) | (0.153) | (0.094) | (0.154) |
| DUA | 0.010 | 0.028 | 0.010 | 0.029 | 0.011 | 0.030 |
| | (0.013) | (0.022) | (0.013) | (0.022) | (0.013) | (0.022) |
| LRO | 0.128 *** | −0.099 *** | 0.126 *** | −0.102 *** | 0.124 *** | −0.095 *** |
| | (0.021) | (0.035) | (0.021) | (0.035) | (0.021) | (0.035) |
| Cons | −0.818 *** | 0.607 *** | −0.813 *** | 0.639 *** | −0.555 *** | 0.474 * |
| | (0.139) | (0.158) | (0.156) | (0.209) | (0.158) | (0.252) |
| N | 9091 | 9091 | 9091 | 9091 | 9091 | 9091 |
| $r^2$ | 0.121 | 0.102 | 0.121 | 0.104 | 0.131 | 0.106 |
| F | 21.72 | 15.26 | 21.09 | 14.63 | 21.27 | 15.23 |
| Hausman_chi2 | 278.34 | 146.45 | 284.66 | 149.66 | 354.42 | 164.45 |
| Time | Yes | Yes | Yes | Yes | Yes | Yes |
| Province | Yes | Yes | Yes | Yes | Yes | Yes |
| Kleibergen–Paap rk LM statistic | - | 167.296 | - | 140.652 | - | 143.920 |
| Cragg–Donald Wald F | - | 296.461 | - | 293.781 | - | 289.048 |

Standard errors in parentheses; *** $p < 0.01$, ** $p < 0.05$, * $p < 0.1$.

## 5. Discussion

This paper studies the impact of the informal supervision mechanism of media attention on green innovation and discusses the regulatory role of environmental regulation and green financial policy. The main findings are as follows:

First, the results of Table 4 indicate that media attention is conductive to improve green innovation. This conclusion is consistent with the research of Chen et al. (2022) [1] and Zhao et al. (2022) [67]. Media pressure resulting from negative media reports can damage the reputation and exterior image of enterprises, which results in limited financing and a decline in enterprise evaluation. This makes the enterprise management have a stronger willingness to reshape the public impression and restore reputation in terms of social corporate responsibility through timely and effective green innovation activities under the pressure portrayed by negative media reports. In addition, different from previous studies, the empirical results also demonstrate that media attention can stimulate both substantive and strategic green innovation.

There is somewhat different from previous research. The results of Table 4 also present that media attention has a positive impact on both substantive and strategic green innovation. It is illustrated that strategic green innovation is conducive to improving the corporate image and reputation in the short term, thus helping enterprises ease the financing constraints and obtain R&D investment, while substantive green innovation is helpful in promoting green technology to reduce the cost of environmental compliance, which

can elevate the economic benefits of enterprises and achieve the sustainable development target in the long term.

Second, the results of Table 5 reveal that environmental regulation has a positive regulatory effect on the relationship between media attention and enterprise green innovation. Moreover, it is also revealed that this regulatory effect is more significant on the effective green innovation. When enterprises attract negative media attention due to environmental problems, environmental regulation constraints implemented by the government becomes stricter, in turn, enterprises have been pushed to fulfill their environmental responsibilities more effectively in addition to preserving the reputation that has been damaged by negative media attention. Formal and informal environmental regulation enables enterprises to focus on environmental issues, which makes the enterprise managers have stronger motivations to reshape their corporate image through effective green technology innovation. Consequently, environmental regulation strengthens the positive impact of media attention on effectiveness of enterprise green innovation.

Third, the results of Table 6 suggest that green finance has a significant positive regulatory effect on the impact of media attention and enterprise green innovation. This implies that the higher the development level of green finance, the greater the financial support for enterprises engaging in green R&D from the government, which can help enterprises under negative media attention obtain financial support in the form of green credit, alleviate their financial constraints, and motivate enterprises to carry out green innovation. Therefore, green finance enhances the positive impact of negative media attention on enterprise green innovation. It also illustrates that the implementation of green finance policy is more effective in improving the substantive green innovation when under the pressure of negative media attention due to environmental problems. The green finance policy implemented by the government pays more attention on the substantive green innovation ability of enterprises, correspondingly, managers are actively in improving the green innovation performance to obtain more innovation benefits and acquire more green finance policy support.

Fourth, from the heterogeneity tests, the results of Table 7 indicate that media attention is more conducive to motivating the green innovation output of heavy polluting enterprises. This is consistent with the conclusion of Zhong and Peng (2022) [68]. Due to the fact that heavily polluting enterprises are subject to strict supervision by the government on the performance of environmental responsibilities and confronted with increasing environmental compliance costs, which makes heavy polluting enterprises have higher enthusiasm to reduce environmental costs through green innovation activities. It is also demonstrated that green innovation is more conducive to stimulating green innovation output of state-owned enterprises. This is consistent with the conclusion of Wang and Zhang (2021) [4]. This may be because state-owned enterprises have a closer relationship with the government and pay more attention to corporate social responsibility performance and social image in the public. Therefore, managers of state-owned enterprises are more active in pursuing green innovation activities to earn more policy supports. Unlike previous studies, heterogeneity test results of regulatory effects illustrate that environmental regulation and green finance are more conducive to boosting the green innovation of heavy pollution enterprises and state-owned enterprises under the pressure from public opinion.

There are still some limitations that need to be discussed in the research. First of all, the concept of green innovation is extensive, including green innovation investment, green innovation output, and efficiency. Based on the widely studied fields, this paper adopts green innovation application and authorization indicators as alternative variables of green innovation, while the impact of media attention on enterprise green innovation investment and green innovation efficiency has not been discussed thoroughly. Consequently, it is necessary to explore and verify the impact of media attention on other enterprise green innovation factors in future research.

There is one more point, the transmission mechanism of media attention to enterprise green innovation needs to be further explored. As a key informal supervision mechanism,

the media plays an important role in green governance. This research analyzes the impact mechanism of media attention on green innovation from the perspective of reputation theory, while does not verify the effectiveness of this transmission mechanism through empirical methods intensively. Correspondingly, how to select reasonable variables to measure the change of enterprise reputation and verify which variables the media focus on affect the enterprises green innovation indirectly are important research areas in the future.

Finally, according to the different functions of environmental regulation tools, environmental regulation can be divided into command-controlled type, market-incentive type, and voluntary type [69]. Based on the emission of environmental pollutants, this research explores the regulatory effect of command-controlled environmental regulation between media attention and green innovation. Nevertheless, how market-incentive type and voluntary type environmental regulation affect media attention and enterprise green innovation has not been involved, which is one of our future research directions. Therefore, it is one of the essential research directions to explore and analyze the heterogeneous impact of diverse environmental regulation tools on media attention and green innovation in the near future.

## 6. Conclusions and Recommendations

This research investigates the impact of media attention on enterprise green innovation under the environmental regulation and green financial policy using data for listed enterprises in China from 2010 to 2019. The results show that media attention is conducive to encourage enterprises to take more active actions in green innovation, and the constraint policy of environmental regulation and the incentive policy of green finance strengthen this impact. The main conclusions are as follows:

(1) Media attention exerted a promotion effect on enterprise green innovation. It has been illustrated that media attention not only improves green innovation, but has a positive impact on both substantive and strategic green innovation.

(2) Environmental regulation has a positive regulatory effect on the relationship between media attention and enterprise green innovation, and this regulatory effect is more significant on the green innovation authorization, while the regulatory effect of environmental regulation in improving substantive and strategic green innovation is no difference for enterprises facing media pressure.

(3) Green finance has a significant positive regulatory effect on the impact of media attention on enterprise green innovation, and this regulatory effect is more significant on the green innovation authorization. The implementation of green finance policy is more effective in improving the substantive green innovation when under the pressure of negative media opinion due to environmental problems.

(4) From the heterogeneity test results, the continuous strengthening of environmental regulation and restriction policies is more conducive to enhancing the media's attention to the green innovation output of state-owned enterprises and heavy polluting enterprises. The positive regulation effect of green financial incentive policy has also reached a similar conclusion.

We believe that our research results highlight the impact of media attention as an informal monitoring tool on enterprise green innovation and provide important enlightenment for policy makers. Based on the above conclusions, recommendations are put forward from the perspectives of government, media, and enterprises.

From the perspective of the media. The media should play an extra-legal supervisory role, drive the mainstream public opinion, and establish a multi governance environmental governance system with government agencies, the public, and other stakeholders. First, the media should further strengthen the role in supervising the implementation of corporate environmental responsibilities, transmit media attention pressure to the enterprises poisoning the environment, especially heavy polluting enterprises and state-owned enterprises, and advocate for green technology innovation by enterprises to reduce the cost of environmental compliance. Second, the media should utilize diverse digital technologies and

intelligent platforms, increase the widely coverage of environmental issues of enterprises, expose the behaviors of enterprises violating environmental responsibilities timely, and enhance the informal supervision role of the media.

From the perspective of the government. First, the government should attach great importance to the supervision and guidance role of the media in the process of implementing green governance, and guide enterprises from passive innovation to active innovation through public opinion publicity and case display. Secondly, the government should strengthen the role of the policy constraints of environmental regulation and raise the policy incentives of green finance in promoting green innovation, guiding enterprises to augment the substantive innovation output. For instance, the government should implement green finance by establishing carbon accounts to enhance the coupling effect of green innovation policies while strengthening environmental regulations.

From the perspective of the enterprises. In the new media era, media has become a key channel for stakeholders to obtain enterprise information. When enterprises attract media attention due to environmental problems, enterprise managers should improve the effectiveness and substance of green innovation to reshape the enterprise image. Therefore, enterprise management should take the initiative to fulfill environmental responsibilities and boost the enterprise's substantive green innovation ability and sustainable development ability effectively.

**Author Contributions:** Data collection, F.W. and H.F.; writing, F.W. and Z.S.; methodology, F.W. and Z.S.; software, F.W.; supervision, Z.S. and H.F. All authors have read and agreed to the published version of the manuscript.

**Funding:** This work was supported by National Social Science Fund Later Funding Project of China (No. 18FGL019, 21FGLB017); National Natural Science Foundation of China (No.L2124032); Jiangsu Province Social Science Fund Major Project (No.22ZDA005); the Project of Philosophy and Social Science Research of Jiangsu Province [2020-10623]; Shandong Social Science Planning Research Project (grant No. 21CGLJ30)); Shandong Social Science Planning Research Project (grant No. 18CKJJ07); Young and Middle-aged Key Teachers Project of Shandong women's University; Humanities and Social Sciences Research Project of Colleges and Universities in Shandong Province(grant No. J17RB120) and Shandong Key Research and Development Program (Soft Science) (grant No. 2021RKY03049).

**Institutional Review Board Statement:** Not applicable.

**Informed Consent Statement:** Not applicable.

**Data Availability Statement:** The data presented in this study are available on request from the corresponding author. The data are not publicly available due to privacy.

**Acknowledgments:** The authors wish to appreciate the valuable comments of the anonymous reviewers. All errors remain the sole responsibility of the authors.

**Conflicts of Interest:** The authors declare no conflict of interest.

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
