# Peer review of "Can Media Attention Promote Green Innovation of Chinese Enterprises? Regulatory Effect of Environmental Regulation and Green Finance"

_sustainability, doi:10.3390/su141711091_

Round 1
Reviewer 1 Report
Please see the comments in the attached file.

Reviewer 2 Report
Dear authors,
First remark is related to the need to adapt the paper title to your case study. You operated with data from China! So, customize accordingly.
Secondly, you have to state clearly the article objectives (better in the Introduction section). Why? This is because to corellate the research hypotheses with the paper objectives.
Meanwhile, check the recommended way of citation in the case of several articles.
Also, explain why you consider as sufficient the number of research hypotheses for your study subject.
Another issue is to explain if figure 1 is designed by you and how this could help an interested reader. This is because in many cases researchers are looking to see the workflow and the logical thread of the article's conception.
In the same time you need to explain why you used in the Variable Definition rather the article of Chinese authors.
Could you mention which type of industries are covered by data sources and what geografically areas are represented by IPC Green Inventory 222 of China's State Patent Office and the World Intellectual Property Organization and CSMAR (maybe here is better to insert a table with these info).
In this subsection (3.2) you inserted several equations which requires numbering (lines 271 to 299). Please check if you described all the notions and elements inserted.
Please describe in the final section your own contribution to the domain, the novelty of the approach, the usefulness of the study related to the possible use in other countries, article limitations.
Reviewer 3 Report
Dear author / s,
thank you very much for the opportunity you have given me to read your manuscript entitled "Does media attention promote enterprise green innovation Based on the regulatory effect of environmental regulation and green finance" submitted for Sustainability Journal. This manuscript discusses about the impact of media attention on enterprise green innovation under the environmental regulation and green financial policy using the data of Chinese listed enterprises from 2010 to 2019 by constructing the fixed effect model.
This is a good work, however there are several concerns need to be resolved. The main concerns regard the originality of the study and the theoretical background, as well as the main implications of the study. These issues must be clear in the whole manuscript, especially in the abstract and introduction. In details, the authors should try to better explain the existent gap in the literature, as well as better motivate the choice of the theory that is supporting this study and the choice of the method over others.
A section entitled to the discussion must be included in the manuscript, as well as a sub-paragraph that highlights the contribution of this study with respect to previous studies on the issues of media attention and enterprise green innovation.
You can best address the issues discussed in your study by also considering the following articles
https://doi.org/10.1016/j.jclepro.2022.132151
https://doi.org/10.1108/JIC-10-2021-0290
In the manuscript you must include a section for academic implications and managerial implications.
I hope you can quickly address every single comment and suggestion included in this report that will help you clarify the contribution of your study and the advancement of knowledge.
Good Luck!
Reviewer 4 Report
The manuscript targets an interesting topic and has publishing potential. However, it needs major improvements in two key areas:
- The method needs better grounding. Your method (dual fixed effects) is not chosen based on adequate preliminary tests for model selection; hence, testing for the necessity of including fixed effects in all models (i.e. with an F test for individual effects) should be accomplished. Additionally, there should be testing for the necessity of including random effects in all models should (i.e. with the Lagrange Multiplier Test for Random Effects). The Hausman test to compare the fixed effects and the random effects models should also be employed. Also, in order to assure that unreliable and spurious regressions are avoided, the unit root properties of the data should be investigated. Finally, in order to address common issues within the fixed effects panel models, namely heteroscedasticity, autocorrelation, and cross-sectional dependence, you should employ commonly used robust standard error computational techniques. Finally, model diagnostics should be included. Errors should be at least tested for serial correlation and for cross-sectional dependence. Otherwise, results are not proven to be robust and you cannot draw reliable inferences from your models.
-There is no Discussion of the results section. Current results should be "discussed" relative to previous findings. This would additionally give the reader a clear understanding of the novelty of the current research.
- The language should be checked and the manuscript proofread. There are many language problems: see for example:"the speed of information dissemination has increasing considerably" - row 31; or "there are still some issues that have not yet been corresponding explored" -row 62
Round 2
Reviewer 2 Report
Dear Authors,
Most of the responses are satisfactory. Still there are needed few more adjustments. For instance, you inserted several equations which requires numbering (lines 281 to 295). Starting with the line 329 all the mathematical equations are numbered. Please keep a unitary way of presentation.
In the same time, you need to reposition the Discussion section before the Conclusions section (this approach is much usual than your approach). Also you need to remove issues and ideas related to the research itself from the Discussion and insert them in the Conclusion (for instance, your own contribution, paper findings etc.).
Finally, please make a check of the paper spelling.
Reviewer 3 Report
Congratulations for the new version of the manuscript. All suggestions and comments have been well addressed.
Good Luck!
Author Response
Dear reviewer,
Thank you for your recognition of the revised paper, which is the motivation for us to continue our follow-up research. At the same time, we also revised the language of the paper, which will help us clearly present the contents of the paper so that readers can better understand it. Thank you again for your hard work and help.
Best regards
Reviewer 4 Report
I appreciate the improved version of your manuscript. There are however further improvements needed. Firstly, the dataset must be more clearly presented. We need to have a clear description of the time and longitudinal dimension of the dataset. Then, all your arguments regarding autocorrelation and stationarity should be included in the paper, and references that back your arguments stated. Otherwise, the reader does not have all the information regarding the reliability of the method. Then, you mention that "In order to avoid unreliability and false regression, this paper uses robust standard error in the model to enhance the reliability of the results in the empirical test". What exactly do you mean by that? what method to adjust the standard errors is implemented that produces heteroskedasticity and autocorrelation consistent errors that are robust to cross-sectional dependence? Subsequently, what tests are accomplished to verify that models are indeed robust? for example, is the Pesaran cross-sectional dependence test performed after model estimations?
These aspects need further work. The language also needs more improvement.
Round 3
Reviewer 4 Report
I find the manuscript further improved. Please take another look at the following aspects, before the paper can be published:
You mentioned that you had added a description of the data characteristics starting at row 339, but you haven't presented the dataset. It is vital for readers to know the number of observations, the no of cross-sections, the time dimension, etc. All these aspects are in turn important and relevant for the method employed.
Then, it is mentioned that robust standard errors are employed (via STATA) to solve the problems of heteroscedasticity and autocorrelation detected. The final step, however, would have been to recheck if the issues are indeed eliminated by this approach. The logical sequence should have been: check the presence of well-known issues- apply robust standard errors- check if the problem has been eliminated. The final step is still not accomplished. Unfortunately, if the problems are still there, then the model is not robust and you cannot make reliable inferences.
Regarding the Discussion of results that has been recommended during the previous review round, there is still some improvement needed. You have included some comparisons with previous studies in section 4.1, but this should be part of the discussion section. The empirical results should be first presented, and afterward, the discussion should reflect how they compare to previous studies, what is different, what is similar, etc.
Round 4
Reviewer 4 Report
I understand your reasoning and arguments. Of course, there is no need to prove the accuracy of the method (i.e. robust standard errors). My recommendation was only to recheck if the issues (auto-corr and heteroscedasticity) are still there after the method is applied or if they have been eliminated. If still there, you can then use the arguments from your response and thus conclude that valid inference can still be made. It is just that the last step (rechecking of the persistence of data issues) is not performed. I maintain that for a full sequence of the implemented data, this step should appear in the manuscript (it would benefit the readers). However, I admit that the arguments introduced in the last version can suffice in the absence of retesting to defend the inference made, thus I do recommend publication. Nonetheless, you can still add the re-testing, I feel that this would increase the value of your manuscript and would satisfy some readers.